# ABCDEG Stress Echocardiography in Aortic Stenosis

**DOI:** 10.3390/diagnostics13101727

**Published:** 2023-05-12

**Authors:** Quirino Ciampi, Lauro Cortigiani, Maria Rivadeneira Ruiz, Andrea Barbieri, Fiore Manganelli, Fabio Mori, Maria Grazia D’Alfonso, Francesca Bursi, Bruno Villari

**Affiliations:** 1Cardiology Division, Fatebenefratelli Hospital, 82100 Benevento, Italy; villari.bruno@gmail.com; 2Cardiology Division, San Luca Hospital, 55100 Lucca, Italy; lacortig@tin.it; 3Cardiology Division, Virgen Macarena University Hospital, 41009 Sevilla, Spain; mariariva.1993@gmail.com; 4Department of Biomedical, Cardiology Division, Metabolic and Neural Sciences, University of Modena and Reggio Emilia, 41121 Modena, Italy; barbieriandrea65@gmail.com; 5Cardiology Division, San Giuseppe Moscati Hospital, 83100 Avellino, Italy; fioremang@gmail.com; 6Cardiology Division, Careggi Hospital, 50134 Florence, Italy; morif@aou-careggi.toscana.it (F.M.); mariagrazia.dalfonso@gmail.com (M.G.D.); 7Department of Health Science, University of Milan, Cardiology Division, San Paolo Hospital, ASST Santi Paolo e Carlo, 20142 Milano, Italy; francescabursi@gmail.com

**Keywords:** aortic stenosis, echocardiography, stress echocardiography, left ventricular contractile reserve, coronary flow reserve

## Abstract

Rest and stress echocardiography (SE) plays a pivotal role in the evaluation of valvular heart disease. The use of SE is recommended in valvular heart disease when there is a mismatch between resting transthoracic echocardiography findings and symptoms. In aortic stenosis (AS), rest echocardiographic analysis is a stepwise approach that begins with the evaluation of aortic valve morphology and proceeds to the measurement of the transvalvular aortic gradient and aortic valve area (AVA) using continuity equations or planimetry. The presence of the following three criteria suggests severe AS: AVA < 1.0 cm^2^, a peak velocity > 4.0 m/s, or a mean gradient > 40 mmHg. However, in approximately one in three cases, we can observe a discordant AVA < 1 cm^2^ with a peak velocity < 4.0 m/s or a mean gradient <40 mmHg. This is due to reduced transvalvular flow associated with LV systolic dysfunction (LVEF < 50%) defined as “classical” low-flow low-gradient (LFLG) AS or normal LVEF “paradoxical” LFLG AS. SE has an established role in evaluating LV contractile reserve (CR) patients with reduced LVEF. In classical LFLG AS, LV CR distinguished pseudo-severe AS from truly severe AS. Some observational data suggest that long-term prognosis in asymptomatic severe AS may not be as favorable as previously thought, offering a window of opportunity for intervention prior to the onset of symptoms. Therefore, guidelines recommend evaluating asymptomatic AS with exercise stress in physically active patients, particularly those younger than 70 years, and symptomatic classical LFLG severe AS with low-dose dobutamine SE. A comprehensive SE assessment includes evaluating valve function (gradients), the global systolic function of the LV, and pulmonary congestion. This assessment integrates considerations of blood pressure response, chronotropic reserve, and symptoms. StressEcho 2030 is a prospective, large-scale study that employs a comprehensive protocol (ABCDEG) to analyze the clinical and echocardiographic phenotypes of AS, capturing various vulnerability sources which support stress echo-driven treatment strategies.

## 1. Introduction

Aortic valve stenosis (AS) is the most common primary valve disease. The disease has a higher incidence of occurrence in the United States and Europe, and a growing prevalence due to the ageing population [1,2].

Stress echocardiography (SE) is an established method for assessing patients with coronary artery disease [3,4]. The comprehensive ABCDE protocol has recently proven itself to be feasible and useful in identifying phenotypes and risk-stratifying patients with chronic coronary syndromes beyond coronary artery stenosis [5]. Conversely, a very low percentage of patients with AS undergo SE in a diagnostic manner to evaluate valve disease diagnosis or severity [6].

The American Society of Echocardiography, the European Association of Cardiovascular Imaging, and the general cardiology guidelines of the American College of Cardiology (ACC), American Heart Association (AHA) and the European Society of Cardiology (ESC) [1,2,7] recognize the role of SE in valvular heart disease and AS, but most recommendations are based on consensus opinions or on small, retrospective studies. Therefore, the versatility of stress echo remains largely unexplored. ABCDE SE may be integrated and enriched to assess transvalvular gradients (step G). However, we need more substantial evidence for a comprehensive evaluation of patients with AS. This is because factors such as inducible ischemia, pulmonary congestion, alterations of preload and LV contractile reserve (CR), coronary microvascular dysfunction and cardiac autonomic balance can be even more critical than valve condition in determining the outcome in valvular heart disease patients. In addition, a targeted therapeutic approach requires a pathophysiological characterization of disease complexity.

The purpose of this review is to evaluate the role of ABCDEG SE in AS patients and, in addition, to analyze the role of SE in specific subgroups: in asymptomatic severe AS patients and in discordant severe AS.

## 2. Methodology

A systematic approach typical to all applications is outlined in the valve SE protocol of execution, as are other parameters must be defined specifically for valvular disease in a certain limelight [8]. With imaging findings, which always require comparison with symptoms, blood pressure response, heart rate and arrhythmias, non-imaging parameters are relevant to the safety and completeness of studies [9].

Patients referred to the SE lab are studied with semi-supine bicycle SE. In patients unable to exercise, a pharmacological test with dobutamine is indicated in selected patients. 

Due to the risk of severe hemodynamic compromise, stress testing is not typically recommended to symptomatic patients with severe AS on account of it being harmful [8,9].

Caution regarding exercise testing is also recommended in cases of AS with peak velocity > 5 m/s or mean gradient > 60 mmHg [10].

Patients with reduced aortic valve area can be classified into four categories, as defined by the guidelines [2]:High-gradient AS;Low-flow low-gradient (LFLG) AS with reduced LVEF;LFLG AS with preserved LVEF;Normal-flow, low-gradient AS with preserved LVEF.

AS patients undergo comprehensive transthoracic echocardiography at rest conditions before SE [11].

AS patients undergo physical or pharmacological SE according to the protocol recommended by the European Association of Cardiovascular Imaging [12] and the American Society of Echocardiography [13].

Severe chest discomfort, diagnostic ST–segment shift, and excessive blood pressure increase, may limit dyspnea, maximal anticipated heart rate, and substantial arrhythmias, all of which are all fundamental grounds for discontinuing the test. During times of stress, there are some basic general rules and acquisition patterns to be considered. The ideal acquisition order, however, should be customized for each patient and distinct research issue. In most cases, the digital capture proceeds in four stages: rest; intermediate stage; peak stress; and recovery (after the conclusion of exercise). Normally, no measurements are required during the image capture phase, while images are saved and examined offline after the performance of the test. This is due to initial expansion of LV end-diastolic volume being a measure of preload reserve and, if impaired, being required to contribute to the decrease in stroke volume and cardiac reserve during periods of stress. Conversely, the intermediate stage is fundamental to assessing changes within a preload reserve.

Concerning B-lines, the recovery stage is decisive. Since pulmonary congestion continues after a stress interruption, B-lines require <1 min acquisition time with a simplified 4-site scan, allowing for information loss during the startling recovery period [14,15,16].

The traditional 2-dimensional echocardiography, the parasternal long-axis view, short-axis view, and the apical 4- and 2-chamber views are deployed in echocardiographic imaging.

All patients underwent Step A. This included an evaluation of anomalies related to wall motion. In a 17-segment model of the left ventricle, the wall motion score index (WMSI), which is a four-point scale ranging from 1 (normal) to 4 (dyskinetic), was determined for every patient involved at baseline and at a time of peak stress [12].

Step B of the methodology involved scoring each site from 0 (normal horizontal A-lines) to 10 (white lung with coalescent B-lines) for B-lines using lung ultrasonography and the 4-site streamlined scan on the third intercostal space [14]. The complex score for all four chest zones varied widely from 0 (all sites with individual site scores of 0) to 40 (all sites with individual site scores of 10). Additionally, there were also four levels of stress B-lines: absence (score points 0–1); mild (2 to 4); moderate (5 to 9); and severe (10 points).

The LV CR was assessed in a force-based fashion in step C of the procedure using the stress/rest ratio of force, a factor was computed as systolic blood pressure/end-systolic volume [17,18].

The left anterior descending coronary artery (LAD) and wall motion were imaged intermittently throughout the conventional SE examination. This was performed with the aim of determining the coronary flow velocity reserve (step D) [19]. Consequently, according to the direction of color Doppler flow mapping, coronary flow in the mid-distal region of the LAD was visualized from modified apical 2- or 3-chamber views. To render offline reviewing and measuring easier, all experiments were digitally stored.

The ratio of the peak assessment to the rest value of a high diastolic flow velocities was used to quantify coronary flow velocity reserve.

Indeed, the peak/rest heart rate ratio from a 12-lead electrocardiogram was used to compute the heart rate reserve (step E) [20,21].

Step G included the evaluation of the trans-aortic maximal and mean gradients using continuous-wave Doppler techniques and the evaluation of integral flow in the LV outflow tract using pulsed-wave Doppler methods. Data stroke volume and AVA were calculated using the continuity equation method. If needed, the steps were repeated after 5 min in the recovery phase (Figure 1).

Beyond SE, adjunctive imaging studies can potentially increase our pathophysiology knowledge of the AS process, contributing additional diagnostic and prognostic information [22]. More specifically, cardiac magnetic resonance outperforms other currently available imaging modalities in identifying, quantifying, and distinguishing between myocardial diseases. Indeed, Rosa et al. [23] provided some insight into the unanswered questions about the mechanism for flow reserve in patients with classical LFLG AS with reduced LVEF during dobutamine SE. They found a lack of association between cardiac magnetic resonance markers of myocardial fibrosis and flow reserve, pointing toward an attenuating but reversible effect of afterload, rather than fibrosis, on myocardial contractility. This hypothesis is consistent with the common scenario in which LVEF improves after aortic valve intervention, often dramatically, despite no flow reserve at baseline [24].

### SE Positivity Criteria

Every positive criterion is predetermined in advance. When two contiguous segments of the same vascular territory of the LV exhibit an increase in at least one point of the segmental score during SE [12], the A criterion is considered to be positive and stand for the existence of stress-induced regional wall motion abnormalities (wall motion score index stress > rest). In the presence of stress or rest B-line 2 units, the B criterion is emphasized as positive [14]. When the force-based LV CR is 2.0 for exercise or dobutamine and 1.1 for vasodilators, the C criterion can be distinguished as positive [17]. When the coronary flow velocity reserve result is less than 2.0, the D criterion is related as positive [19]. On the other hand, when heart rate reserve is present, the E criterion is perceived favorably in the presence of heart rate reserve < 1.80 for exercise or dobutamine or <1.22 for dipyridamole or adenosine [20,21]. The G criteria include the stroke volume increase of >20%, indicative of significant LV CR [9], and an increase in AVA (>1 cm^2^), indicative of pseudo-severe AS, and a lack of increase in AVA, indicative of true severe AS. During the last decade, the evidence supporting the optimal cut-off for the trans-aortic gradient augmentation has not increased as expected. Therefore, major scientific research needs to evaluate the positivity criteria cut-off of this parameter.

## 3. Stress Echo in Specific Subgroups of AS

### 3.1. Stress Echo in Asymptomatic AS

Some observational data suggest that long-term prognosis in asymptomatic AS stenosis may not be as favorable as previously thought, suggesting that there exists a window of opportunity for intervention prior to the onset of symptoms. Recent data indicate that there is an increased risk of mortality in asymptomatic severe AS, even when LVEF is >50% [25]. Moreover, Banovic et al. [26] in the AVATAR trial demonstrated that, in asymptomatic severe AS, early surgery aortic valve replacement reduced the occurrence spontaneous follow-up events (death, acute myocardial infarction, stroke, acute heart failure) compared with conservative treatment, regardless of the symptoms concerned.

When exercise testing causes symptoms, the patient is deemed to be symptomatic and qualifies for an aortic valve replacement class 1 recommendation. Symptoms are considered symptoms if they are reported deliberately by the patient or detected by exercise testing. Normally, in patients without overt symptoms who exhibit (1) a drop of >10 mm Hg in systolic blood pressure from baseline to peak exercise or (2) a significant decrease in exercise tolerance compared to age and sex normal standards, the rate of symptom onset is elevated within the bounds of 1 to 2 years (roughly 60% to 80%) [27].

Therefore, guidelines recommend evaluating asymptomatic AS with exercise stress in physically active patients, particularly those younger than 70 years. A decreased exercise tolerance indicates intervention for the 2020 ACC/AHA [1] and 2021 ESC guidelines [2].

The 2021 ESC recommendations state that, if the resting LVEF is lower than 55% without evidence of an additional cause, intervention should be considered in patients with severe AS who are asymptomatic. Additionally, this is advised if the LVEF is less than 50% (class of recommendation 1) [2]. Given that underestimating symptoms is common in AS and that patients often limit their level of activity over time to cope with the disease’s slow progression, stress testing combined with exercise is helpful in identifying patients who are truly asymptomatic.

The incidence of the abnormal stress test in asymptomatic severe AS has ranged from 28% to 67%, with a pooled average of 49% [28]. A lower % of age–sex–predicted metabolic equivalents and slower heart rate recovery are associated with long-term mortality. If patients prove to have a preserved exercise capacity, a safe deferral of surgery is proposed for the next 1 year [29].

Therefore, recording abnormal aortic valve hemodynamics with exercise echocardiography is of limited additive value and is no longer recommended for decision making.

The predictive value of the test is diminished in older adults despite the fact that a negative exercise test result may seem comforting in younger patients. Combining them with the echocardiographic evaluation of LV function, transvalvular pressure gradients and pulmonary arterial pressure can be useful.

In this case, the application of ABCDE SE in asymptomatic AS patients has the potential to identify the presence of latent coronary artery disease (Step A+); the presence of B lines at peak stress (Step B+), indicative of a preclinical decompensation which may precede the onset of dyspnea and symptoms by days or months; a latent LV dysfunction with reduced LV CR (Step C+); and an impaired coronary flow velocity reserve (Step D+). This is particularly the case in in patients who have angiographically normal coronary arteries and severe AS. When a valve is replaced prior to the regression of LV hypertrophy in these patients, the degree of coronary microvascular disease, which is reflected in a reduction in coronary flow reserve during stress testing and is related to exercise capacity, is reversible and foresees outcomes better than the severity of AS [30,31,32,33,34]. Therefore, it is well established that coronary microcirculation is impaired in AS, impacting myocardial remodeling, aortic flow patterns, and clinical progression [35].

There are invasive and more reproducible invasive tests to assess absolute coronary flow and coronary flow reserve in patients with severe AS candidate to TAVI/SAVR. In patients with severe AS and non-obstructive coronary artery disease, an invasive study demonstrated that the compensatory mechanism of increased resting flow maintains adequate perfusion at rest, but not during hyperemia, with a consequent reduction in coronary flow reserve [36].

The management of patients with moderate AS is very important, particularly in terms of the choice between early intervention versus watchful waiting. The correct diagnosis of true moderate AS is of primary importance. For an accurate measurement of the trans-aortic jet velocity, multiple acoustic windows should be used in order to determine the highest velocity, In fact, alignment errors in the Doppler beam led to an underestimation of the true velocity, and consequently to the calculated gradients underestimating AS severity. In addition, an echocardiographic evaluation of valve morphology and aortic valve calcium scoring can be performed using computer tomography (CT) to confirm the diagnosis of true moderate AS. Paolisso et at. [37] demonstrated that the heart valve clinic approach, within a multidisciplinary team, is associated with better management in moderate AS in relation to diagnostic tests, education, and treatment. Additionally, this approach was shown to be an independent predictor of reduced all-cause death in patients with moderate AS.

### 3.2. Stress Echo in Discordant Severe AS

The evaluation of classical LFLG AS is the main indication of SE in valvular heart disease [38]. Classical LFLG AS is defined as a mean gradient < 40 mmHg, valve area ≤ 1 cm^2^, LV EF < 50%, and stroke volume index ≤ 35 mL/m^2^.

Due to the difficulty of separating patients with severe AS from those who have pseudo-severe AS, this entity poses a diagnostic challenge [1,2,9,39]. The small area of the aortic valve plays a role in the severity of AS by increasing afterload, decreasing LVEF, and decreasing cardiac output. In pseudo-severe AS, the severity of the condition is overstated because the underlying LV systolic dysfunction reduces the opening force, causing an incomplete valve opening [40]. In patients with classical LFLG AS, low-dose dobutamine SE (up to 20 mcg/kg/min) is recommended by ESC guidelines (class 1), and it may be helpful to determine the trans-valvular gradient and the AVA to differentiate severe (true AS) from pseudo-severe AS [9,13,39,40,41].

The primary objective of using dobutamine SE is to boost the transvalvular flow rate without causing myocardial ischemia. In response to an increase in transvalvular flow rate, patients with pseudo-severe AS will have an increase in the AVA and little change in gradient, whereas patients with severe AS will possess a fixed valve area but undergo an increase in stroke volume and gradient [2,9,13,39,40]. Therefore, it is considered reasonable to use low dose dobutamine SE to define severity further and assess CR with the ACC/AHA 2020 guidelines [1].

Evaluation of the LV CR is a critical component of classical LF-LG AS because patients with reduced LV CR have a higher risk of unfavorable events [9,13,39]. The imaging assessment is based on the assessment of flow reserve (an increase in stroke volume of at least 20%) and LV systolic function (changes in EF or global longitudinal strain) [9,13,39], an analysis of changes in pressure gradients, and AVA (Figure 2). An increase in stroke volume >20% denotes significant LV CR [39,40].

An imbalance between the severity of the stenosis and myocardial reserve, an inadequate increase in myocardial blood flow because of associated coronary artery disease, irreversible myocardial damage from a previous myocardial infarction or extensive myocardial fibrosis can all contribute to the lack of stroke volume increase during dobutamine SE [38].

A noninvasive method of assessing LV CR can be inferred to add another modality [41,42,43] because changes in LV force are a load-independent measure of left ventricular contractility. The enhancement in LVEF is typically used to assess LV CR. However, it depends on heart rate, preload, afterload, synchrony of contraction, and other factors to combined to generate myocardial contractility, which may occasionally even be deceptive [42].

The LV CR measured by LV force, however, significantly varies from LVEF when the situation is viewed from a conceptual, methodological, and clinical perspective. It only requires the measurement of systolic blood pressure by cuff sphygmomanometer and end-systolic volume by 2D echocardiography, rather than the assessment end-diastolic and end-systolic volume (Figure 3). It is independent of preload and afterload changes [43], which instead affect LVEF.

Additionally, with all modalities of SE, LV CR is more prognostically effective than LVEF changes in identifying patients at higher risk, including both those with normal and those with noticeably abnormal resting LV function [44,45,46]. Systolic blood pressure should be added to the peak AV gradient in AS patients to evaluate LV force. Stress Echo 2030, a prospective multicenter study designed to collect direct information on the safety, viability, and outcomes of a multiparametric approach to different subsets of pathophysiological conditions besides coronary artery disease, such as valvular heart disease [47], can be used to test out this parameter.

Patients with an increased indexed stroke volume of at least 20%, an unchanged AVA, and an unchanged transvalvular gradient have indeterminate AS severity. Up to 30% of patients lack CR, indicating a high risk of perioperative mortality following surgical aortic valve replacement [48,49]. Projected AVA (AVAProj) at a typical transvalvular flow rate has been suggested as a new parameter with which to obtain this significant limitation.

This parameter’s goal is to predict the AVA would be at a standard normal flow rate of 250 mL/s. When compared to conventional DSE parameters, it has been discovered that in patients with LG-AS, the AVAProj more accurately foresees underlying AS severity, the impairment of myocardial blood flow, LV pump reserve, and survival [50]. Thus, the AVAProj has the potential to elevate the diagnostic precision of DSE to being able to differentiate between severe and pseudosevere AS.

In discordant severe AS, the confirmation of the actual AS severity is essential. Both the American and European guidelines recommend performing non-contrast CT to quantitate aortic valve calcification and therefore differentiate true severe vs. non-severe AS, and this is particularly true in the case of patients with paradoxically low flow and LG-AS. Additionally, CT cut-offs have been defined to identify patients who likely have an AVA <1 cm^2^ [1,2]. However, at present, no prospective studies have used CT calcium scoring to help identify true severe AS amongst a cohort in which the results of CT findings guided aortic valve intervention. Furthermore, the reported experience of a CT approach in patients with low-gradient AS with impaired LVEF is relatively small (fewer than 200 individuals in total), and the value of a ‘negative’ CT calcium score within this cohort is unclear. Hence, further studies are needed to determine the outcome and impact of AVR in patients with discordant severe AS according to flow (low vs. normal) and confirmed AS severity (true severe vs. non-severe) [51].

### 3.3. Stress Echo in AS and Prognosis

In patients with classical LF-LG AS, severe impairment of LV longitudinal systolic function at rest or during dobutamine SE, very low LVEF (35%), and the absence of LV CR are the main factors that have been linked to an increased risk of mortality under conservative management, as well as following aortic valve replacement [9,13,48,49,50].

SE has an important diagnostic and prognostic role in evaluating LV CR patients with heart failure patients and reducing LV EF [52,53].

The prognostic role of LV CR for AS patients who are candidates for transfemoral aortic valve implantation (TAVI) or surgical aortic valve replacement (SAVR) remains to be clarified [51,52].

In LF-LG AS patients undergoing SAVR, the peri-procedural risk is considered greater if LV CR is absent [41].

Patients with LV CR have a significantly better results with percutaneous or surgical aortic valve replacement than with medical therapy in LF-LG traditional severe AS [41,42,49,50]. Aortic valve replacement surgery has a high operative mortality rate (6–33%) and can be associated with the absence of LV CR during dobutamine SE in about one-third of patients [49,50]. This factor does not, however, predict the absence of LV function improvement, the overcoming of symptomatic status, or late survival afterwards surgery [49,50].

However, the outcome and management of classical LFLG AS has changed: the operative outcome of SAVR has improved in LF-LG AS, with much lower operative mortality and less frequent severe prosthesis–patient mismatch [54,55,56]. TAVI has emerged as a valuable alternative to SAVR due to the less invasive nature of the procedure, the lower requirements of postoperative care, and the superior toleration of the percutaneous procedure altogether by the patient [55,57].

The prognostic role of SE in LF-LG AS remains uncertain and needs to be clarified with further scientific evidence. Ribeiro et al. [58] showed that the absence of LV CR was not associated with any adverse effect on clinical outcomes or LVEF changes at follow-up. The same results were found by Buchanan et al. [59], who did not find differences in all-cause mortality in the presence or absence of LV CR in LF-LG AS; Sato et al. [24] and Maes et al. [60] confirmed that the absence of LV CR had no effect on clinical outcomes or changes in LVEF over time. Annabi et al. [61] demonstrated that in patients with classical LFLG AS produced excellent outcomes at one year following TAVI, regardless of the presence or absence of LV CR at preprocedural dobutamine SE.

Conversely, when working on LFLG AS patients undergoing TAVI, Barbash et al. [62] showed that CR that was assessed with dobutamine SE did not predict LVEF recovery but was associated with lower mortality [21]. Similarly, Hayek et al. [63] showed that in LFLG AS patients without LV CR had worse survival than those with LV CR. Finally, D’Andrea et al. [64] showed that LV CR, assessed by global longitudinal strain during dobutamine SE, was associated with significant LV reverse remodeling after the TAVI procedure.

Based on this evidence, the current ESC guidelines [2] suggest that intervention should be considered in LF-LG AS patients with reduced LVEF without LV CR, mainly when cardiac CT calcium scoring confirms severe aortic stenosis with a class of recommendation IIa; for LF-LG AS patients with reduced LVEF and evidence of LV CR, ESC guidelines suggest intervention with a higher class of recommendation (Ib).

AS is a common valvular disease, especially affecting the elderly. As for atrial fibrillation, the incidences of this issue increase with age. Many studies in different subgroups of AS have shown that atrial fibrillation is associated with worse outcomes [65,66,67]. In fact, atrial fibrillation is a marker of underlying structural abnormalities: LV hypertrophy, LV diastolic dysfunction, and left atrial dilatation, each of which can be associated with atrial fibrillation [67,68]. Atrial fibrillation, precipitating symptom onset, may play an important role in the clinical course of the AS [68].

Finally, AS should be considered in the bleeding risk assessment of patients with atrial fibrillation with an increased risk of gastrointestinal bleeding [69].

## 4. Conclusions

SE, when combined with exercise or dobutamine, has essential applications in the assessment of patients with AS. Its potential is, however, underused [6,70].

In asymptomatic patients with AS, it is possible to have the presence of LV hypertrophy LV systolic dysfunction, left atrial enlargement, mitral and tricuspid regurgitation, and right ventricular dysfunction. Aortic valve replacement does not affect prognosis in asymptomatic patients; only after the patient develops symptoms is a benefit obtained with aortic valve replacement since it improves the outcomes of these issues. However, a new and increasingly accredited hypothesis supports the concept that cardiac damage identifies patients with poor prognosis, despite the absence of symptoms, when imaging shows parameters of cardiac damage related to poor outcome [23].

An innovative concept of staging AS, centered on the patient rather than imaging, has been proposed in order to assist with the decision to potentially intervene more quickly This is an alternative to simply categorizing the severity of the valve lesion based on echocardiographic data [71].

In this framework, the comprehensive approach with the ABCDEG protocol may refine the AS phenotype identification and it may play an important role in its practical use: in fact, ABCDEG SE contributes to the risk stratification of AS patients characterized by numerous vulnerabilities beyond coronary artery stenosis [5]: myocardial ischemia (Step A), pulmonary congestion (Step B), left ventricular CR (Step C), coronary microcirculatory reserve (Step D), sympathetic cardiac autonomic reserve (Step E), and transvalvular gradients (Step G). This thorough SE method might work well for determining the variety of clinical phenotypes, prognostic weaknesses, and potential therapies related to difficult patients [72].

## Figures and Tables

**Figure 1 diagnostics-13-01727-f001:**
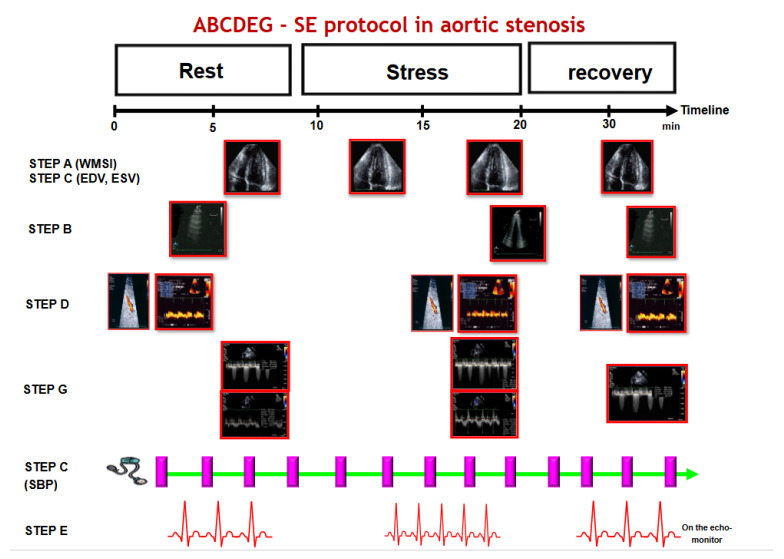
The ABCDEG protocol in AS. The entire test takes roughly 30 min to be completed, including the preparation phase and the written response. Step A and step C require the very same images. Step E requires only one electrocardiogram lead. Step B can be completed before starting the SE and in the early recovery phase. Step D requires peak diastolic Doppler evaluation of the mid-distal part of the left anterior descending coronary artery; evaluations should be performed at rest and peak stress. Step G requires the maximal and mean trans-aortic gradient and the integral flow in the outflow tract and involves assessment at rest and at peak stress. Modified from Ciampi Q et al. [3].

**Figure 2 diagnostics-13-01727-f002:**
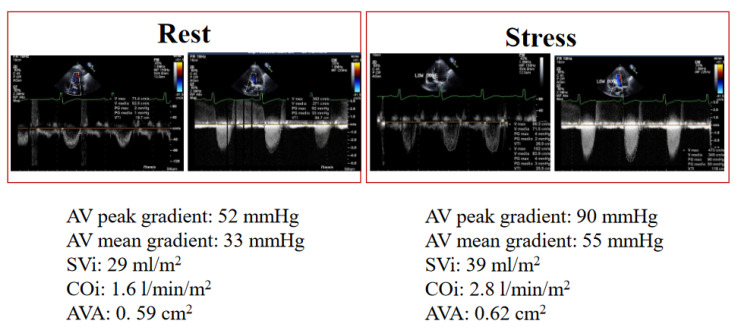
Rest and at peak stress of low doses of dobutamine SE. SVi: stroke volume index; COi: cardiac output index; AVA: aortic valve area.

**Figure 3 diagnostics-13-01727-f003:**
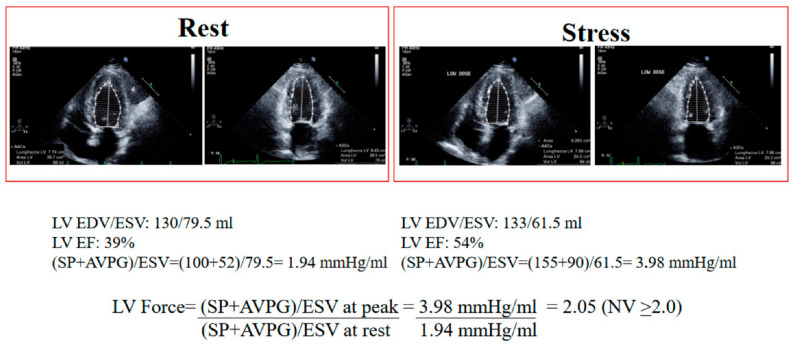
Rest and at peak stress of low-dose of dobutamine SE. EDV: end-diastolic volume; ESV: end-systolic volume; LV EF: left ventricular ejection fraction; SBP: systolic blood pressure; AVPG: aortic valve peak gradient; NV: normal value.

## Data Availability

No new data was created in this review. This is a literature review of previously published data, reported in the references.

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
