# Peer review of "ABCDEG Stress Echocardiography in Aortic Stenosis"

_diagnostics, 2023, doi:10.3390/diagnostics13101727_

Round 1

Reviewer 1 Report

Ciampi et al. discussed the value of rest and stress echocardiography (SE) in patients with aortic stenosis (AS). Of interest, the comprehensive approach with the ABCDE protocol may re-fine the AS phenotype identification and contribute to the risk stratification of patients characterized by numerous vulnerabilities beyond coronary artery stenosis. This SE approach might work well for capturing various clinical phenotypes, prognostic weaknesses, and potential therapies for difficult patients.

Globally, the review is well written, and the authors should be commended for this exhaustive and well-organized manuscript; nevertheless, I have some concerns that need to be addressed before it can be re-submitted.

1)    Please discuss the impact/limitation of atrial fibrillation on SE in patients with AS.

2)    Please integrate the Section related to the treatment and outcomes of patients with moderate and asymptomatic severe AS with the following references (PMID: 34779220, 36598073).

3)    What is the role of calcium score assessed by cardiac CT in pts with LFLG AS?

4)    Page 4, paragraphs regarding coronary flow and CFR and pag.5 lines 206-211: please specify that nowadays, there are invasive and more reproducible invasive tests to assess absolute coronary flow and CFR in patients with severe AS candidate to TAVR/SAVR; in patients with severe AS and non-obstructive coronary artery disease, with the progression of LV hypertrophy, the compensatory mechanism of increased resting flow maintains adequate perfusion at rest, but not during hyperemia. As a consequence, both CFR and MRR are significantly impaired (PMID: 35977812).

Minor editing of English language required

Author Response

Reviewer #1: Thank you for your thoughtful and constructive criticism in bold your comments; in red font parts added to the revised manuscript.

Ciampi et al. discussed the value of rest and stress echocardiography (SE) in patients with aortic stenosis (AS). Of interest, the comprehensive approach with the ABCDE protocol may re-fine the AS phenotype identification and contribute to the risk stratification of patients characterized by numerous vulnerabilities beyond coronary artery stenosis. This SE approach might work well for capturing various clinical phenotypes, prognostic weaknesses, and potential therapies for difficult patients.

Globally, the review is well written, and the authors should be commended for this exhaustive and well-organized manuscript; nevertheless, I have some concerns that need to be addressed before it can be re-submitted.

1)    Please discuss the impact/limitation of atrial fibrillation on SE in patients with AS.

Many thanks for this important question: In the section Stress echo in AS and prognosis we add: “AS is a common valvular disease affecting especially the elderly. As for atrial fibrillation the incidences increase with age. Many studies in different subgroups of AS have shown that AF is associated with worse outcomes (66-68). In fact, atrial fibrillation is a marker of underlying structural abnormalities: LV hypertrophy, LV diastolic dysfunction, and left atrial dilatation, which can be associated with atrial fibrillation (68-69). Atrial fibrillation precipitating symptom onset, may play an important role in the clinical course of the AS (69).

Finally, AS should be considered in the bleeding risk assessment of patients with atrial fibrillation with an increased risk of gastrointestinal bleeding (70).

We add the new references #66-70

2)    Please integrate the Section related to the treatment and outcomes of patients with moderate and asymptomatic severe AS with the following references (PMID: 34779220, 36598073).

 In the section Stress echo in asymptomatic AS we add: “Moreover, Banovic et al (26) in the AVATAR trial demonstrated that in asymptomatic severe AS, early surgery aortic valve replacement reduced spontaneous follow-up events (death, acute myocardial infarction, stroke, acute heart failure) compared with conservative treatment, regardless the symptoms.

We add the new reference #26

In the section Stress echo in asymptomatic AS we add: The management of patients with moderate AS is very important, particularly the choice between early intervention versus watchful waiting. The right diagnosis of true moderate AS is primary: for an accurate measurement of the trans-aortic jet velocity, multiple acoustic windows should be used in order to determine the highest velocity: in fact, alignment errors in Doppler beam lead to an underestimation of the true velocity, and consequently of the calculated gradients resulting in an underestimation of AS severity. In addition, echocardiographic evaluation of valve morphology and aortic valve calcium scoring using computer tomography (CT) to confirm the diagnosis of true moderate AS. Paolisso et at (37) demonstrated that heart valve clinics approach, within a multidisciplinary team, was associated with better management in moderate AS for diagnostic tests, education and treatment and this approach was an independent predictor of reduced all-cause death in patients with moderate AS.

We add the new reference #37

3)    What is the role of calcium score assessed by cardiac CT in pts with LFLG AS?

In the section Stress echo in discordant severe AS, we add “In discordant severe AS, confirmation of the actual AS severity is essential. Both the American and European guidelines recommend performing non-contrast CT to quantitate aortic valve calcification and therefore differentiate true-severe vs. non-severe AS, particularly in patients with paradoxical low-flow, LG-AS, and CT cut-offs have been defined to identify patients who likely have an AVA <1 cm2 (1-2). However, currently, no prospective studies have used CT calcium scoring to help identify true-severe AS amongst this cohort in which the results of CT findings guided aortic valve intervention. Furthermore, the reported experience of a CT approach in patients with low-gradient AS with impaired LVEF is relatively small (fewer than 200 individuals in total), and the value of a ‘negative’ CT calcium score within this cohort is unclear. Hence, further studies are needed to determine the outcome and impact of AVR in patients with discordant severe AS, according to flow (low vs. normal) and confirmed AS severity (true-severe vs. non-severe) (51).

We add the new reference #51

4)    Page 4, paragraphs regarding coronary flow and CFR and pag.5 lines 206-211: please specify that nowadays, there are invasive and more reproducible invasive tests to assess absolute coronary flow and CFR in patients with severe AS candidate to TAVR/SAVR; in patients with severe AS and non-obstructive coronary artery disease, with the progression of LV hypertrophy, the compensatory mechanism of increased resting flow maintains adequate perfusion at rest, but not during hyperemia. As a consequence, both CFR and MRR are significantly impaired (PMID: 35977812).

As reviewer suggests, we add in the revised manuscript: “There are invasive and more reproducible invasive tests to assess absolute coronary flow and coronary flow reserve in patients with severe AS candidate to TAVI/SAVR; in patients with severe AS and non-obstructive coronary artery disease, an invasive study demonstrated that the compensatory mechanism of increased resting flow maintains adequate perfusion at rest, but not during hyperemia, with consequent reduction in coronary flow reserve (36).

We add the new reference #36

Reviewer 2 Report

I reviewed with interest the manuscript of Ciampi et al. "ABCDEG Stress Echocardiography in Aortic Stenosis". In this review, the authors consider the possibility of using the ABCDEG Stress Echocardiography protocol to clarify indications for valve surgery in patients with aortic stenosis. In addition, the authors propose to supplement the existing ABCDEG Stress Echocardiography protocol, which is used in patients with chronic coronary syndrome, with additional indicators. Among them, the authors include: assessment of regurgitant flows (step F), transvalvular gradients (step G), left atrial volume (step L), pulmonary pressures (step P), and right ventricular function (step R). As a result, it is planned to use the ABCDEG+ protocol. Apparently, this protocol will improve the diagnosis in patients with aortic stenosis, will clarify the indications for surgery and, ultimately, improve the prognosis.

Nevertheless, when reviewing the manuscript, I had questions and comments that I would like to receive answers from the authors.

1. The authors did not formulate a clear purpose of the review, this is required.

2. Some of the links are wrong. Thus, reference 3 (Contemporary presentation and management of valvular heart disease: the EURObservational research program valvular heart disease II survey) cannot refer to the phrase "The comprehensive ABCDE protocol, has recently proved to be feasible and useful in identifying phenotypes and risk-stratifying patients with chronic coronary syndromes beyond coronary artery stenosis." (lines 45-47). Accordingly, reference 4 (Prognostic value of ABCDE stress echocardiography) does not match the phrase "Conversely, a very low percentage of patients with aortic valve stenosis (AS) underwent SE in the diagnostic way to evaluate valve disease diagnosis or severity" (lines 47-49 ).

2. The caption to figure 1 contains repetitions (lines 139-144): "The entire test takes roughly 30 minutes to be completed, including the preparation phase and the written response. If the test is positive, the recovery period is recommended but optional. Only one ECG lead is required for Step E. Step B can be accomplished in the first 30 seconds of recuperation. the written response takes about 30'. The recovery phase is optional and advised in the presence of test positivity. Step A and step C require the very same images. Step E requires only one electrocardiogram lead. Step B can be completed in the early ( < 30 sec) recovery phase".

3. For me, the question remained unclear - the authors already propose the ABCDEG + protocol for practical use, or they just substantiate the upcoming scientific studies on its validation (for example, Stress Echo 2030). Authors should state this more clearly in their review.

4. What surprised me was the names of the steps of the extended ABCDEG+ protocol (assessment of regurgitant flows (step F), transvalvular gradients (step G), left atrial volume (step L), pulmonary pressures (step P), and right ventricular function (step R )) appeared only at the end of the review. At the same time, the review does not provide justification for the names of the stages and the need to evaluate them in order to solve diagnostic and tactical approaches to the management of patients with aortic stenosis.

5. Information after the text of the article (Author Contributions and Data Availability Statement) needs to be corrected.

No comments

Author Response

Reviewer #2: Thank you for your thoughtful and constructive criticism in bold your comments; in red font parts added to the revised manuscript.

I reviewed with interest the manuscript of Ciampi et al. "ABCDEG Stress Echocardiography in Aortic Stenosis". In this review, the authors consider the possibility of using the ABCDEG Stress Echocardiography protocol to clarify indications for valve surgery in patients with aortic stenosis. In addition, the authors propose to supplement the existing ABCDEG Stress Echocardiography protocol, which is used in patients with chronic coronary syndrome, with additional indicators. Among them, the authors include: assessment of regurgitant flows (step F), transvalvular gradients (step G), left atrial volume (step L), pulmonary pressures (step P), and right ventricular function (step R). As a result, it is planned to use the ABCDEG+ protocol. Apparently, this protocol will improve the diagnosis in patients with aortic stenosis, will clarify the indications for surgery and, ultimately, improve the prognosis.

Nevertheless, when reviewing the manuscript, I had questions and comments that I would like to receive answers from the authors.

  1. The authors did not formulate a clear purpose of the review, this is required.

In the introduction section of the revised manuscript we add, as suggested: The propose of this review is to evaluate the role of ABCDEG SE in AS patients and, in addition, to analyze the role of SE in specific subgroups: asymptomatic severe AS patients and in discordant severe AS.

  1. Some of the links are wrong. Thus, reference 3 (Contemporary presentation and management of valvular heart disease: the EURObservational research program valvular heart disease II survey) cannot refer to the phrase "The comprehensive ABCDE protocol, has recently proved to be feasible and useful in identifying phenotypes and risk-stratifying patients with chronic coronary syndromes beyond coronary artery stenosis." (lines 45-47). Accordingly, reference 4 (Prognostic value of ABCDE stress echocardiography) does not match the phrase "Conversely, a very low percentage of patients with aortic valve stenosis (AS) underwent SE in the diagnostic way to evaluate valve disease diagnosis or severity" (lines 47-49 )

Thanks to underline the mistake in the number of references. I changed reference #3 and #4

  1. The caption to figure 1 contains repetitions (lines 139-144): "The entire test takes roughly 30 minutes to be completed, including the preparation phase and the written response. If the test is positive, the recovery period is recommended but optional. Only one ECG lead is required for Step E. Step B can be accomplished in the first 30 seconds of recuperation. the written response takes about 30'. The recovery phase is optional and advised in the presence of test positivity. Step A and step C require the very same images. Step E requires only one electrocardiogram lead. Step B can be completed in the early ( < 30 sec) recovery phase".

Many thanks. We changed the legend of the Figure 1: “The entire test takes roughly 30 minutes to be completed, including the preparation phase and the written response. Step A and step C require the very same images. Step E requires only one electrocardiogram lead. Step B can be completed before starting the SE and in the early recovery phase. Step D requires peak diastolic Doppler evaluation of the mid-distal part of the left anterior descending coronary artery, evaluated at rest and peak stress. Step G requires the maximal and mean trans-aortic gradient and the integral flow in the outflow tract and assessed at rest and at peak stress. Modified from Ciampi Q et al (3)”

  1. For me, the question remained unclear - the authors already propose the ABCDEG + protocol for practical use, or they just substantiate the upcoming scientific studies on its validation (for example, Stress Echo 2030). Authors should state this more clearly in their review.

Many thanks for the question. We propose ABCDEG in clinical practice for the study of patients with aortic stenosis. In the discussion section we add: “In this framework, the comprehensive approach with the ABCDE protocol may refine the AS phenotype identification and it may play an important role in the practical use: in fact, ABCDEG SE contributes in the risk stratification of AS patients characterized by numerous vulnerabilities…”

  1. What surprised me was the names of the steps of the extended ABCDEG+ protocol (assessment of regurgitant flows (step F), transvalvular gradients (step G), left atrial volume (step L), pulmonary pressures (step P), and right ventricular function (step R )) appeared only at the end of the review. At the same time, the review does not provide justification for the names of the stages and the need to evaluate them in order to solve diagnostic and tactical approaches to the management of patients with aortic stenosis.

I agree with your criticism. I removed in the revised manuscript the sentence in the conclusion section: In AS, the assessment of regurgitant flows (step F), transvalvular gradients (step G), left atrial volume (step L), pulmonary pressures (step P), and right ventricular function (step R) is integrated and enriched (ABCDE +) as necessary into the core ABCDE protocol.

In addition, in the introduction section, I changed the sentence: ABCDE SE may be integrated and enriched to assess regurgitant flows, transvalvular gradients (step G), left atrial volume, pulmonary pressures, and right ventricular function.

A comprehensive SE assessment includes evaluating valve function (gradients and regurgitation), the global systolic and diastolic function of the left ventricle (LV), left atrial volume, and pulmonary congestion, pulmonary arterial pressure, and right ventricular function.

However, we need more substantial evidence for a comprehensive evaluation of patients with AS, as factors such as inducible ischemia, pulmonary congestion, alterations of preload and LV contractile reserve (CR), coronary microvascular dysfunction, and cardiac autonomic balance, left and right atrial volume changes, pulmonary hemodynamic response, and right ventricular function reserve can be even more critical than valve condition in determining the outcome in valvular heart disease patients.

  1. Information after the text of the article (Author Contributions and Data Availability Statement) needs to be corrected.

I Correct the Author Contributions and Data Availability Statement in the revised version of the manuscript.

Author Contributions: For research articles with several authors, the following statements should be used “Conceptualization, Q.C.and L.C.; methodology, F.B..; software, MRC, MGDA.; validation, B.V. A.B. F.M..; formal analysis, F.B.; investigation, L.C..; resources, A.B.; data curation, MGDA.; writing—original draft preparation, Q.C..; writing—review and editing, Q.C., A.B. and F.M.; visualization, Q.C., L.C..; supervision, B.V..; project administration, B.V..; funding acquisition, none. All authors have read and agreed to the published version of the manuscript.”

Data Availability Statement: Data Availability Statement: No new data was created in this review. This is a literature review of previously published data, reported in the references.

Reviewer 3 Report

This is a well-written, comprehensive manuscript on an important diagnostic problem.

I only have 2 suggestions:

The introduction should be supplemented with epidemiological data that would emphasize the importance of the problem of aortic stenosis.

Due to the development of advanced imaging methods, the discussion should discuss the potential possibilities of performing functional tests in this type of patients using computed tomography and magnetic resonance imaging. The advantages of echocardiography over these methods, but also the limitations of echocardiography that these methods remove, should be discussed.

Author Response

Reviewer #3 Thank you for your thoughtful and constructive criticism in bold your comments; in red font parts added to the revised manuscript.

This is a well-written, comprehensive manuscript on an important diagnostic problem. I only have 2 suggestions:

The introduction should be supplemented with epidemiological data that would emphasize the importance of the problem of aortic stenosis.

Many thanks for your comment. In the introduction section we add:  “Aortic stenosis (AS) is the most common primary valve disease, with higher incidence of in the United States and Europe, and a growing prevalence due to the ageing population (1-2)”

Due to the development of advanced imaging methods, the discussion should discuss the potential possibilities of performing functional tests in this type of patients using computed tomography and magnetic resonance imaging. The advantages of echocardiography over these methods, but also the limitations of echocardiography that these methods remove, should be discussed.

In the methodology section we add: “Beyond SE, adjunctive imaging studies can potentially increase our pathophysiology knowledge of the AS process, contributing additional diagnostic and prognostic information (22). More specifically, cardiac magnetic resonance outperforms other currently available imaging modalities in identifying, quantifying, and distinguishing between myocardial diseases. Indeed, Rosa et al. (23) provided some insight into the unanswered questions about the mechanism for flow reserve in patients with classical LF-LG severe aortic stenosis during dobutamine SE.  They found a lack of association between cardiac magnetic resonance markers of myocardial fibrosis and flow reserve pointing toward an attenuating but reversible effect of afterload, rather than fibrosis, on myocardial contractility. This hypothesis is consistent with the common scenario where EF improves after aortic valve intervention, often dramatically, despite no flow reserve at baseline (24).

We add 3 new references #22-23-24

Reviewer 4 Report

Thank you very much for an opportunity to read and evaluated very interesting manuscript written by Ciampi Q et al.

The paper presents a systematic review of ABCDEG protocol – Stress Echocardiography in Aortic Stenosis patients.  That new data shows that SE has changed in last years. The new protocol will be used in selected patients by additional parameters such as G for intraventricular and trans ventricular gradients. 

SE (exercise and dobutamine) is an important tool in evaluating of valvular heart disease especially aortic stenosis. Authors suggest that not many patients underwent SE in the diagnostic way and evaluation of aortic stenosis severity.

 It’s a systematic review for patients with AS divided into four groups as defined by guidelines and underwent Step A, B,C,D  and G. The ABCDEG protocol in AS is described with details and include three phase (rest, stress and recover) which depends on results. 

The methodology also describes SE positivity criteria. Indications for stress echo in specific subgroups of AS were also analyzed following by ESC recommendations. The significance of SE in assessing patients is potential but not used. 

Several limitations were also included in study. I find this article very interesting and relevant. The study should be continued in the future. It will be interesting to find results with multicenter studies.

Author Response

Reviewer #4

Thank you very much for an opportunity to read and evaluated very interesting manuscript written by Ciampi Q et al.

The paper presents a systematic review of ABCDEG protocol – Stress Echocardiography in Aortic Stenosis patients.  That new data shows that SE has changed in last years. The new protocol will be used in selected patients by additional parameters such as G for intraventricular and trans ventricular gradients. 

SE (exercise and dobutamine) is an important tool in evaluating of valvular heart disease especially aortic stenosis. Authors suggest that not many patients underwent SE in the diagnostic way and evaluation of aortic stenosis severity.

 It’s a systematic review for patients with AS divided into four groups as defined by guidelines and underwent Step A, B,C,D  and G. The ABCDEG protocol in AS is described with details and include three phase (rest, stress and recover) which depends on results. 

The methodology also describes SE positivity criteria. Indications for stress echo in specific subgroups of AS were also analyzed following by ESC recommendations. The significance of SE in assessing patients is potential but not used. 

Several limitations were also included in study. I find this article very interesting and relevant. The study should be continued in the future. It will be interesting to find results with multicenter studies.

Many thanks for your comments

Round 2

Reviewer 1 Report

The authors address all the issues raised by the reviewer. No more strong remarks from my side.

Reviewer 2 Report

The authors have responded to my comments and amended the text. I have no other comments.

No comments